# *SDXL:* Improving Latent Diffusion Models for High-Resolution Image Synthesis

**Dustin Podell**    **Zion English**    **Kyle Lacey**    **Andreas Blattmann**    **Tim Dockhorn**

**Jonas Müller**                **Joe Penna**                **Robin Rombach**

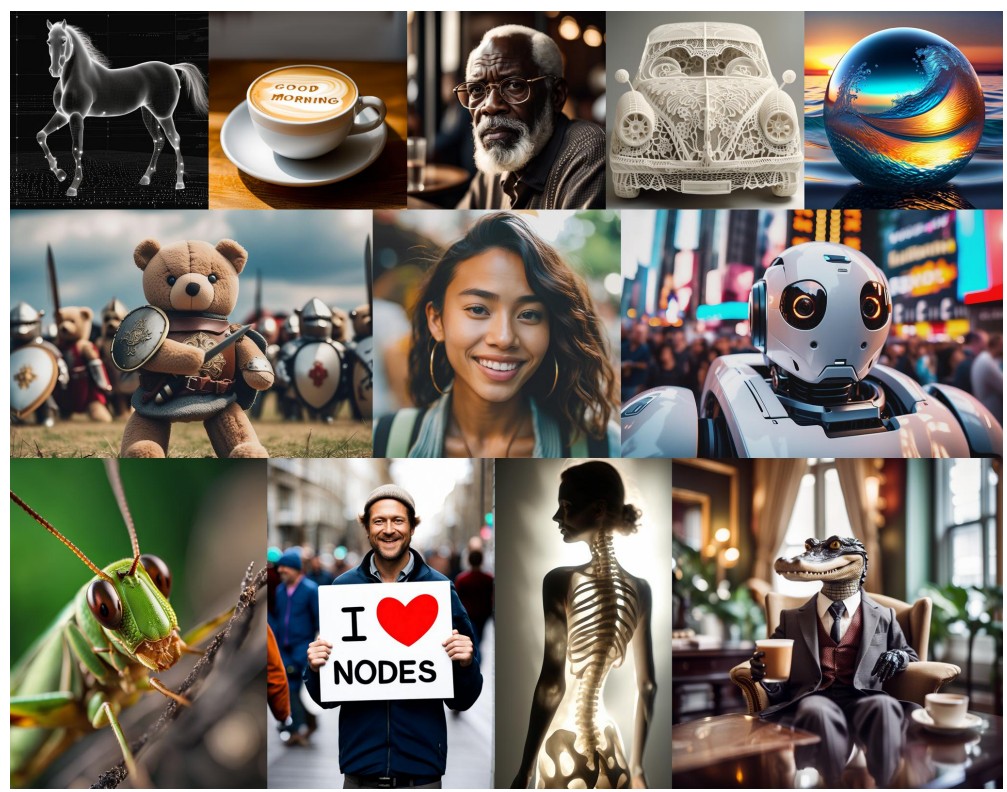

## Abstract

We present *Stable Diffusion XL* (*SDXL*), a latent diffusion model for text-to-image synthesis. Compared to previous versions of *Stable Diffusion*, *SDXL* leverages a three times larger UNet backbone, achieved by significantly increasing the number of attention blocks and including a second text encoder. Further, we design multiple novel conditioning schemes and train *SDXL* on multiple aspect ratios. To ensure highest quality results, we also introduce a *refinement model* which is used to improve the visual fidelity of samples generated by *SDXL* using a post-hoc *image-to-image* technique. We demonstrate that *SDXL* improves dramatically over previous versions of *Stable Diffusion* and achieves results competitive with those of black-box state-of-the-art image generators such as Midjourney (Holz, 2023).

## 1 Introduction

The last year has brought enormous leaps in deep generative modeling across various data domains, such as natural language (Touvron et al., 2023), audio (Huang et al., 2023), and visual media (Rombach et al., 2021; Ramesh et al., 2022; Saharia et al., 2022; Singer et al., 2022; Ho et al., 2022; Blattmann et al., 2023; Esser et al., 2023). In this report, we focus on the latter and unveil *SDXL*, a drastically improved version of *Stable Diffusion*. *Stable Diffusion* is a latent text-to-image diffusion model (DM) which serves as the foundation for an array of recent advancements in, e.g.,

3D classification (Shen et al., 2023), controllable image editing (Zhang & Agrawala, 2023), image personalization (Gal et al., 2022), synthetic data augmentation (Stöckl, 2022), graphical user interface prototyping (Wei et al., 2023), etc. Remarkably, the scope of applications has been extraordinarily extensive, encompassing fields as diverse as music generation (Forsgren & Martiros, 2022) and reconstructing images from fMRI brain scans (Takagi & Nishimoto, 2023).

User studies demonstrate that *SDXL* consistently surpasses all previous versions of *Stable Diffusion* by a significant margin (see Fig. 1). In this report, we present the design choices which lead to this boost in performance encompassing *i)* a 3× larger UNet-backbone compared to previous *Stable Diffusion* models (Sec. 2.1), *ii)* two simple yet effective additional conditioning techniques (Sec. 2.2) which do not require any form of additional supervision, and *iii)* a separate diffusion-based refinement model which applies a noising-denoising process (Meng et al., 2021) to the latents produced by *SDXL* to improve the visual quality of its samples (Sec. 2.5).

A major concern in the field of visual media creation is that while black-box-models are often recognized as state-of-the-art, the opacity of their architecture prevents faithfully assessing and validating their performance. This lack of transparency hampers reproducibility, stifles innovation, and prevents the community from building upon these models to further the progress of science and art. Moreover, these closed-source strategies make it challenging to assess the biases and limitations of these models in an impartial and objective way, which is crucial for their responsible and ethical deployment. With *SDXL* we are releasing an *open* model that achieves competitive performance with black-box image generation models (see Fig. 11 & Fig. 12).

## 2 IMPROVING *Stable Diffusion*

In this section we present our improvements for the *Stable Diffusion* architecture. These are modular, and can be used individually or together to extend any model. Although the following strategies are implemented as extensions to latent diffusion models (LDMs) (Rombach et al., 2021), most of them are also applicable to their pixel-space counterparts.

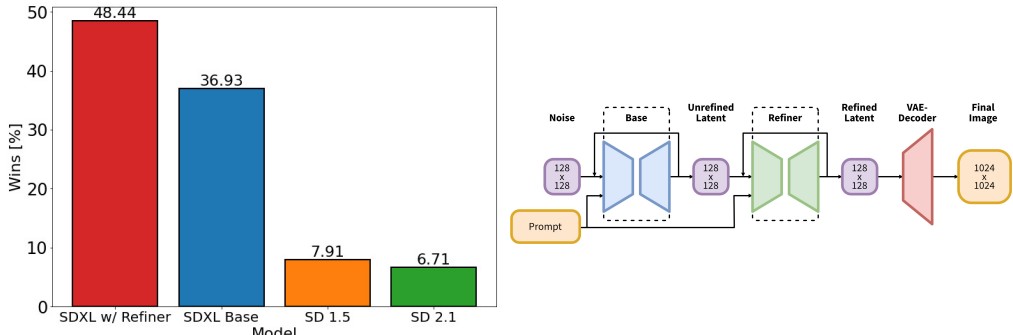

Figure 1: *Left:* Comparing user preferences between *SDXL* and *Stable Diffusion* 1.5 & 2.1. While *SDXL* already clearly outperforms *Stable Diffusion* 1.5 & 2.1, adding the additional refinement stage boosts performance. *Right:* Visualization of the two-stage pipeline: We generate initial latents of size 128 × 128 using *SDXL*. Afterwards, we utilize a specialized high-resolution *refinement model* and apply SDEdit (Meng et al., 2021) on the latents generated in the first step, using the same prompt. *SDXL* and the refinement model use the same autoencoder.

### 2.1 ARCHITECTURE & SCALE

Starting with the seminal works Ho et al. (2020) and Song et al. (2020b), which demonstrated that DMs are powerful generative models for image synthesis, the convolutional UNet (Ronneberger et al., 2015) architecture has been the dominant architecture for diffusion-based image synthesis. However, with the development of foundational DMs (Saharia et al., 2022; Ramesh et al., 2022; Rombach et al., 2021), the underlying architecture has constantly evolved: from adding self-attention and improved upscaling layers (Dhariwal & Nichol, 2021), over cross-attention for text-to-image synthesis (Rombach et al., 2021), to pure transformer-based architectures (Peebles & Xie, 2022).

Table 1: Comparison of *SDXL* and older *Stable Diffusion* models.

| Model | *SDXL* | SD 1.4/1.5 | SD 2.0/2.1 |
|---|---|---|---|
| # of UNet params | 2.6B | 860M | 865M |
| Transformer blocks | [0, 2, 10] | [1, 1, 1, 1] | [1, 1, 1, 1] |
| Channel mult. | [1, 2, 4] | [1, 2, 4, 4] | [1, 2, 4, 4] |
| Text encoder | CLIP ViT-L & OpenCLIP ViT-bigG | CLIP ViT-L | OpenCLIP ViT-H |
| Context dim. | 2048 | 768 | 1024 |
| Pooled text emb. | OpenCLIP ViT-bigG | N/A | N/A |

We follow this trend and, following Hoogeboom et al. (2023), shift the bulk of the transformer computation to lower-level features in the UNet. In particular, and in contrast to the original *Stable Diffusion* architecture, we use a heterogeneous distribution of transformer blocks within the UNet: For efficiency reasons, we omit the transformer block at the highest feature level, use 2 and 10 blocks at the lower levels, and remove the lowest level ($8\times$ downsampling) in the UNet altogether — see Tab. 1 for a comparison between the architectures of *Stable Diffusion* 1.x & 2.x and *SDXL*. We opt for a more powerful pre-trained text encoder that we use for text conditioning. Specifically, we use OpenCLIP ViT-bigG (Ilharco et al., 2021) in combination with CLIP ViT-L (Radford et al., 2021), where we concatenate the penultimate text encoder outputs along the channel-axis (Balaji et al., 2022). Besides using cross-attention layers to condition the model on the text-input, we follow Nichol et al. (2021) and additionally condition the model on the pooled text embedding from the OpenCLIP model. These changes result in a model size of 2.6B parameters in the UNet, see Tab. 1. The text encoders have a total size of 817M parameters.

## 2.2 MICRO-CONDITIONING

**Conditioning the Model on Image Size** A notorious shortcoming of the LDM paradigm (Rombach et al., 2021) is the fact that training a model requires a *minimal image size*, due to its two-stage architecture. The two main approaches to tackle this problem are either to discard all training images below a certain minimal resolution (for example, *Stable Diffusion* 1.4/1.5 discarded all images with any size below 512 pixels), or, alternatively, upscale images that are too small. However, depending on the desired image resolution, the former method can lead to significant portions of the training data being discarded, what will likely lead to a loss in performance and hurt generalization. We visualize such effects in Fig. 2 for the dataset on which *SDXL* was pretrained. For this particular choice of data, discarding all samples below our pretraining resolution of $256^2$ pixels would lead to a significant 39% of discarded data. The second method, on

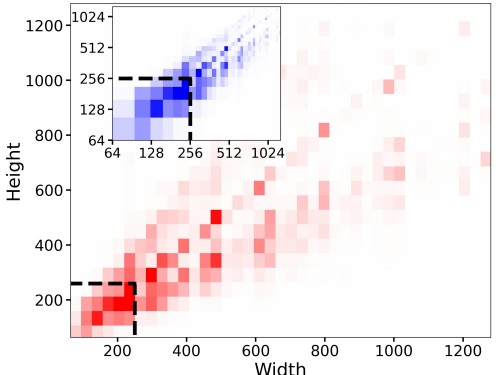

Figure 2: Height-vs-Width distribution of our pre-training dataset. Without the proposed size-conditioning, 39% of the data would be discarded due to edge lengths smaller than 256 pixels as visualized by the dashed black lines. Color intensity in each visualized cell is proportional to the number of samples.

the other hand, usually introduces upscaling artifacts which may leak into the final model outputs, causing, for example, blurry samples.

Instead, we propose to condition the UNet model on the original image resolution, which is trivially available during training. In particular, we provide the original (i.e., before any rescaling) height and width of the images as an additional conditioning to the model $\mathbf{c}_{\text{size}} = (h_{\text{original}}, w_{\text{original}})$. Each component is independently embedded using a Fourier feature encoding, and these encodings are concatenated into a single vector that we feed into the model by adding it to the timestep embedding (Dhariwal & Nichol, 2021).

At inference time, a user can then set the desired *apparent resolution* of the image via this *size-conditioning*. Evidently (see Fig. 3), the model has learned to associate the conditioning $c_{\text{size}}$ with resolution-dependent image features, which can be leveraged to modify the appearance of an output corresponding to a given prompt. Note that for the visualization shown in Fig. 3, we visualize samples

| $\mathbf{c}_{\text{size}} = (64, 64)$ | $\mathbf{c}_{\text{size}} = (128, 128),$ | $\mathbf{c}_{\text{size}} = (256, 256),$ | $\mathbf{c}_{\text{size}} = (512, 512),$ |
|---|---|---|---|

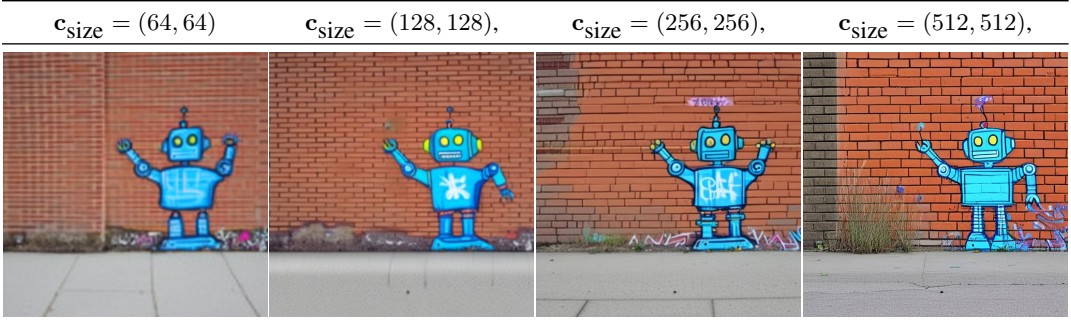

*"A robot painted as graffiti on a brick wall. a sidewalk is in front of the wall, and grass is growing out of cracks in the concrete."*

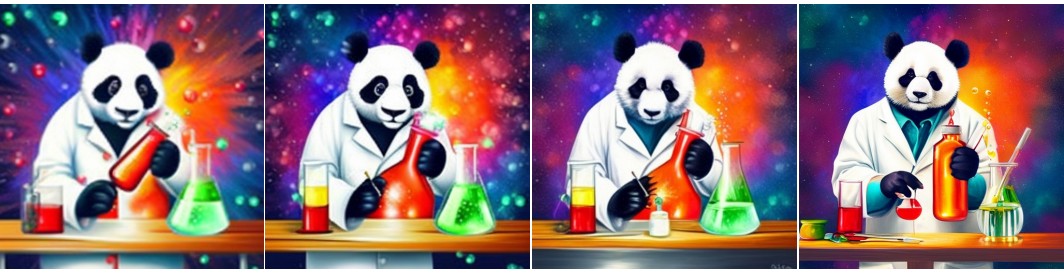

*"Panda mad scientist mixing sparkling chemicals, artstation."*

Figure 3: The effects of varying the size-conditioning: We show draw 4 samples with the same random seed from *SDXL* and vary the size-conditioning as depicted above each column. The image quality clearly increases when conditioning on larger image sizes. Samples from the $512^2$ model, see Sec. 2.5. Note: For this visualization, we use the $512 \times 512$ pixel base model (see Sec. 2.5), since the effect of size conditioning is more clearly visible before $1024 \times 1024$ finetuning. Best viewed zoomed in.

generated by the $512 \times 512$ model (see Sec. 2.5 for details), since the effects of the size conditioning are less clearly visible after the subsequent multi-aspect (ratio) finetuning which we use for our final *SDXL* model.

We quantitatively assess the effects of this simple but effective conditioning technique by training and evaluating three LDMs on class conditional ImageNet (Deng et al., 2009) at spatial size $512^2$: For the first model (*CIN-512-only*) we discard all training examples with at least one edge smaller than $512$ pixels what results in a train dataset of only 70k images. For *CIN-nocond* we use all training examples but without size conditioning. This additional conditioning is only used for *CIN-size-cond*. After training we generate 5k samples with 50 DDIM steps (Song et al.,

Table 2: Conditioning on the original spatial size of the training examples improves performance on class-conditional ImageNet Deng et al. (2009) on $512^2$ resolution.

| model | FID-5k $\downarrow$ | IS-5k $\uparrow$ |
|---|---|---|
| *CIN-512-only* | 43.84 | 110.64 |
| *CIN-nocond* | 39.76 | 211.50 |
| *CIN-size-cond* | **36.53** | **215.34** |

2020a) and (classifier-free) guidance scale of 5 (Ho & Salimans, 2022) for every model and compute IS Salimans et al. (2016) and FID Heusel et al. (2017) (against the full validation set). For *CIN-size-cond* we generate samples always conditioned on $\mathbf{c}_{\text{size}} = (512, 512)$. Tab. 2 summarizes the results and verifies that *CIN-size-cond* improves upon the baseline models in both metrics. We attribute the degraded performance of *CIN-512-only* to bad generalization due to overfitting on the small training dataset while the effects of a mode of blurry samples in the sample distribution of *CIN-nocond* result in a reduced FID score. Note that, although we find these classical quantitative scores not to be suitable for evaluating the performance of foundational (text-to-image) DMs Saharia et al. (2022); Ramesh et al. (2022); Rombach et al. (2021) (see App. E), they remain reasonable metrics on ImageNet as the neural backbones of FID and IS have been trained on ImageNet itself.

**Conditioning the Model on Cropping Parameters** The first two rows of Fig. 4 illustrate a typical failure mode of previous *SD* models: Synthesized objects can be cropped, such as the cut-off head of the cat in the left examples for *SD* 1-5 and *SD* 2-1. An intuitive explanation for this behavior is the use of *random cropping* during training of the model: As collating a batch in DL frameworks such as PyTorch (Paszke et al., 2019) requires tensors of the same size, a typical processing pipeline

| | *"A propaganda poster depicting a cat dressed as french emperor napoleon holding a piece of cheese."* | *"a close-up of a fire spitting dragon, cinematic shot."* |

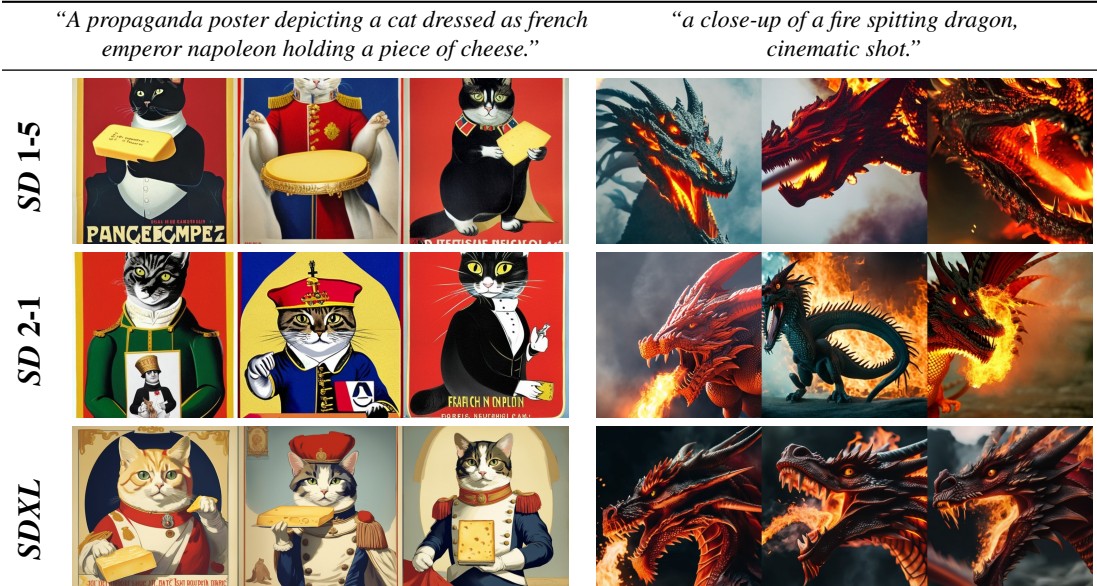

Figure 4: Comparison of the output of *SDXL* with previous versions of *Stable Diffusion*. For each prompt, we show 3 random samples of the respective model for 50 steps of the DDIM sampler Song et al. (2020a) and cfg-scale 8.0 Ho & Salimans (2022). Additional samples in Fig. 15.

is to (i) resize an image such that the shortest size matches the desired target size, followed by (ii) randomly cropping the image along the longer axis. While random cropping is a natural form of data augmentation, it can leak into the generated samples, causing the malicious effects shown above.

To fix this problem, we propose another simple yet effective conditioning method: During dataloading, we uniformly sample crop coordinates $c_{\text{top}}$ and $c_{\text{left}}$ (integers specifying the amount of pixels cropped from the top-left corner along the height and width axes, respectively) and feed them into the model as conditioning parameters via Fourier feature embeddings, similar to the size conditioning described above. The concatenated embedding $\mathbf{c}_{\text{crop}}$ is then used as an additional conditioning parameter. We emphasize that this technique is not limited to LDMs and could be used for any DM. Note that crop- and size-conditioning can be readily combined. In such a case, we concatenate the feature embedding along the channel dimension, before adding it to the timestep embedding in the UNet. Alg. 1 illustrates how we sample $\mathbf{c}_{\text{crop}}$ and $\mathbf{c}_{\text{size}}$ during training if such a combination is applied.

Given that in our experience large scale datasets are, on average, object-centric, we set $(c_{\text{top}}, c_{\text{left}}) = (0, 0)$ during inference and thereby obtain object-centered samples from the trained model.

See Fig. 5 for an illustration: By tuning $(c_{\text{top}}, c_{\text{left}})$, we can successfully *simulate* the amount of cropping during inference. This is a form of *conditioning-augmentation*, and has been used in various forms with AR (Jun et al., 2020) models, and recently with diffusion models (Karras et al., 2022).

While other methods such as "data bucketing" (NovelAI, 2023) successfully tackle the same task, we still benefit from cropping-induced data augmentation, while making sure that it does not leak into the generation process - we actually use it to our advantage to gain more control over the image synthesis process. Furthermore, it is easy to implement and can

---

**Algorithm 1** Size- and crop-micro-conditioning

**Require:** Training dataset of images $\mathcal{D}$
**Require:** Target image size for training $\boldsymbol{s} = (h_{\text{tgt}}, w_{\text{tgt}})$
**Require:** Resizing function $\boldsymbol{R}$
**Require:** cropping function function $\boldsymbol{C}$
**Require:** Model train step $\boldsymbol{T}$
  converged ← False
  **while** not converged **do**
    $x \sim \mathcal{D}$
    $w_{\text{original}} \leftarrow \text{width}(x)$
    $h_{\text{original}} \leftarrow \text{height}(x)$
    $\mathbf{c}_{\text{size}} \leftarrow (h_{\text{original}}, w_{\text{original}})$
    $x \leftarrow \boldsymbol{R}(x, \boldsymbol{s})$   ▷ resize smaller image size to target size $\boldsymbol{s}$
    **if** $h_{\text{original}} \leq w_{\text{original}}$ **then**
      $c_{\text{left}} \sim \mathcal{U}(0, \text{width}(x) - s_w)$   ▷ sample $c_{\text{left}}$
      $c_{\text{top}} = 0$
    **else if** $h_{\text{original}} > w_{\text{original}}$ **then**
      $c_{\text{top}} \sim \mathcal{U}(0, \text{height}(x) - s_h)$   ▷ sample $c_{\text{top}}$
      $c_{\text{left}} = 0$
    **end if**
    $\mathbf{c}_{\text{crop}} \leftarrow (c_{\text{top}}, c_{\text{left}})$
    $x \leftarrow \boldsymbol{C}(x, \boldsymbol{s}, \mathbf{c}_{\text{crop}})$   ▷ crop image to size $\boldsymbol{s}$
    converged ← $\boldsymbol{T}(x, \mathbf{c}_{\text{size}}, \mathbf{c}_{\text{crop}})$   ▷ train model
  **end while**

| $\mathbf{c}_{\text{crop}} = (0,0)$ | $\mathbf{c}_{\text{crop}} = (0,256),$ | $\mathbf{c}_{\text{crop}} = (256,0),$ | $\mathbf{c}_{\text{crop}} = (512,512),$ |
| --- | --- | --- | --- |

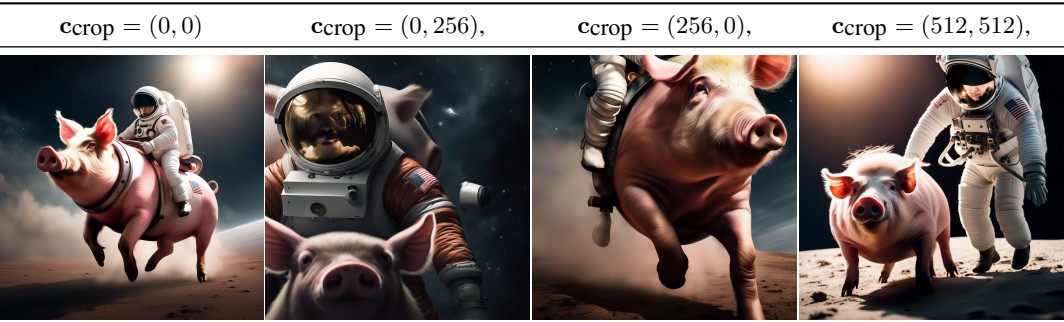

*"An astronaut riding a pig, highly realistic dslr photo, cinematic shot."*

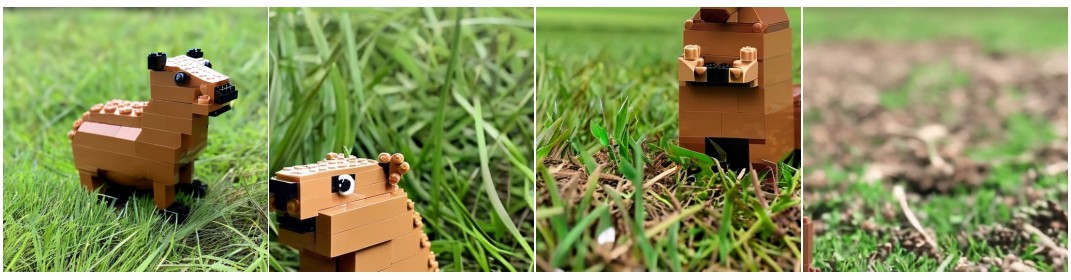

*"A capybara made of lego sitting in a realistic, natural field."*

Figure 5: Varying the crop conditioning as discussed in Sec. 2.2. See Fig. 4 and Fig. 15 for samples from *SD* 1.5 and *SD* 2.1 which provide no explicit control of this parameter and thus introduce cropping artifacts. Samples from the $512^2$ model, see Sec. 2.5.

be applied in an online fashion during training, without additional data preprocessing.

## 2.3 MULTI-ASPECT TRAINING

Real-world datasets include images of widely varying sizes and aspect-ratios (c.f. fig. 2) While the common output resolutions for text-to-image models are square images of $512 \times 512$ or $1024 \times 1024$ pixels, we argue that this is a rather unnatural choice, given the widespread distribution and use of landscape (e.g., 16:9) or portrait format screens.

Motivated by this observation, we finetune our model to handle multiple aspect-ratios simultaneously: We follow common practice (NovelAI, 2023) and partition the data into buckets of different aspect ratios, where we keep the pixel count as close to $1024^2$ pixels as possibly, varying height and width accordingly in multiples of 64. A full list of all aspect ratios used for training is provided in App. H. During optimization, a training batch is composed of images from the same bucket, and we alternate between bucket sizes for each training step. Additionally, the model receives the bucket size (or, *target size*) as a conditioning, represented as a tuple of integers $\mathbf{c}_{\text{ar}} = (h_{\text{tgt}}, w_{\text{tgt}})$ which are embedded into a Fourier space similarly to the size- and crop-conditionings described above.

In practice, we apply multi-aspect training as a finetuning stage after pretraining the model at a fixed aspect-ratio and resolution and combine it with the conditioning techniques introduced in Sec. 2.2 via concatenation along the channel axis. Fig. 17 in App. I provides `python`-code for this operation. Note that crop-conditioning and multi-aspect training are complementary operations, and crop-conditioning then only works within the bucket boundaries (usually 64 pixels). For ease of implementation, however, we opt to keep this control parameter for multi-aspect models. We note that this mechanism can be extended to joint *multi-aspect, multi-resolution* training by varying the total pixel density. In practice, this means that the batch size can be adjusted dynamically depending on the current resolution, to make the best use of availalable VRAM.

## 2.4 IMPROVED AUTOENCODER

*Stable Diffusion* is a *LDM*, operating in a pretrained, learned (and fixed) latent space of an autoencoder (AE). While the bulk of the semantic composition is done by the LDM (Rombach et al., 2021), we can improve *local*, high-frequency details in generated images by improving the AE. To this end, we train the same AE architecture used for the original *Stable Diffusion* at a batch-size of 256 and additionally track the weights with an exponential moving average. The resulting AE outperforms the original model in all evaluated reconstruction metrics, see Tab. 3. A small ablation for the influence of these parameters is reported in App. J, we find that EMA is helpful in all our settings, while the effects of the large batch size are mixed. We use this AE for all of our experiments.

Table 3: Autoencoder reconstruction performance on the COCO2017 Lin et al. (2015) validation split, images of size $256 \times 256$ pixels. Note: *Stable Diffusion* 2.x uses an improved version of *Stable Diffusion* 1.x's autoencoder, where the decoder was finetuned with a reduced weight on the perceptual loss Zhang et al. (2018), and used more compute. Note that our new autoencoder is trained from scratch.

| model | PNSR ↑ | SSIM ↑ | LPIPS ↓ | rFID ↓ |
|---|---|---|---|---|
| *SDXL*-VAE | **24.7** | **0.73** | **0.88** | **4.4** |
| *SD*-VAE 1.x | 23.4 | 0.69 | 0.96 | 5.0 |
| *SD*-VAE 2.x | 24.5 | 0.71 | 0.92 | 4.7 |

## 2.5 PUTTING EVERYTHING TOGETHER

We train the final model, *SDXL*, in a multi-stage procedure. *SDXL* uses the autoencoder from Sec. 2.4 and a discrete-time diffusion schedule (Ho et al., 2020; Sohl-Dickstein et al., 2015) with 1000 steps. First, we pretrain a base model (see Tab. 1) on an internal dataset whose height- and width-distribution is visualized in Fig. 2 for 600 000 optimization steps at a resolution of $256 \times 256$ pixels and a batch-size of 2048, using size- and crop-conditioning as described in Sec. 2.2. We continue training on 512 px for another 200 000 optimization steps, and finally utilize multi-aspect training (Sec. 2.3) in combination with an offset-noise (Guttenberg & CrossLabs, 2023; Lin et al., 2023) level of 0.05 to train the model on different aspect ratios (Sec. 2.3, App. H) of $\sim 1024 \times 1024$ pixel area.

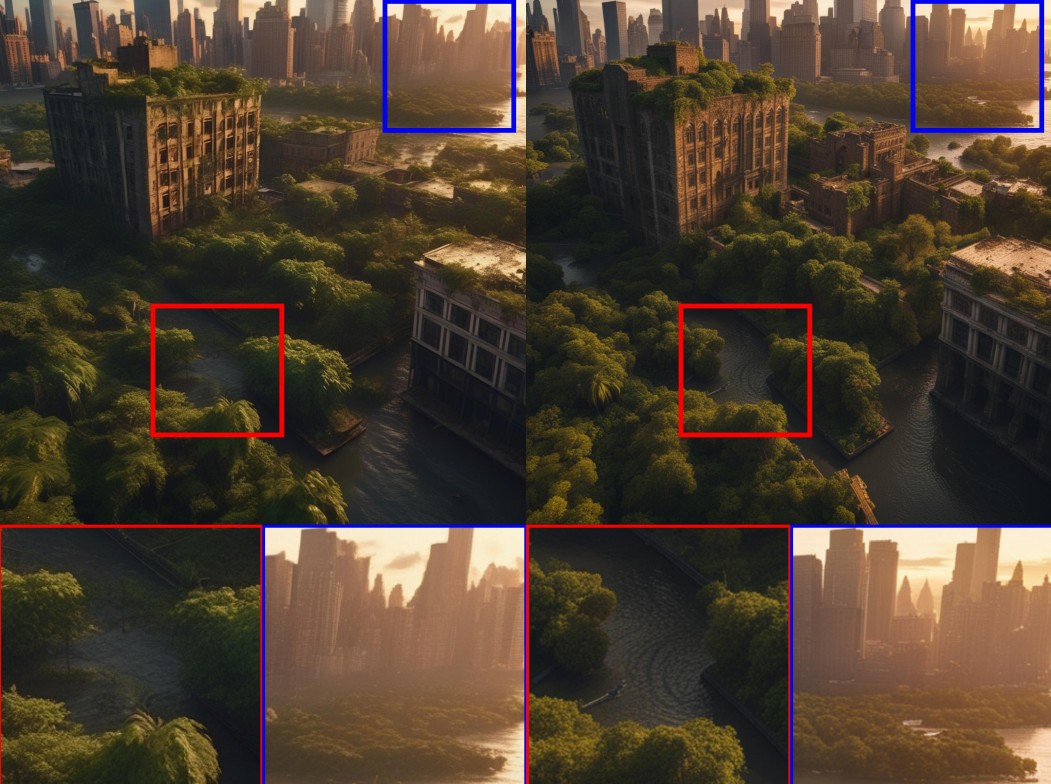

Figure 6: $1024^2$ samples (with zoom-ins) from *SDXL* without (left) and with (right) the **refiner model** (see Sec. 2.5). Prompt: *"Epic long distance cityscape photo of New York City flooded by the ocean and overgrown buildings and jungle ruins in rainforest, at sunset, cinematic shot, highly detailed, 8k, golden light"*. See Fig. 14 for additional samples.

**Refinement Stage**   Empirically, we find that the resulting model sometimes yields samples of low local quality, see Fig. 6. To improve sample quality, we train a separate LDM in the same latent space, which is specialized on high-quality, high resolution data and employ a noising-denoising process as introduced by *SDEdit* (Meng et al., 2021) on the samples from the base model, or, alternatively, finish the denoising process with the refiner. We follow (Balaji et al., 2022) and specialize this refinement model on the first 200 (discrete) noise scales. During inference, we render latents from the base *SDXL*, and directly diffuse and denoise them in latent space with the refinement model (see Fig. 1), using the same text input. We note that this step is optional, but improves sample quality for detailed backgrounds and human faces, as demonstrated in Fig. 6 and Fig. 14.

To assess the performance of our model (with and without refinement stage), we conduct a user study, and let users pick their favorite generation from the following four models: *SDXL*, *SDXL* (with refiner), *Stable Diffusion* 1.5 and *Stable Diffusion* 2.1. The results demonstrate the *SDXL* with the refinement stage is the highest rated choice, and outperforms *Stable Diffusion* 1.5 & 2.1 by a significant margin (win rates: *SDXL* w/ refinement: $48.44\%$, *SDXL* base: $36.93\%$, *Stable Diffusion* 1.5: $7.91\%$, *Stable Diffusion* 2.1: $6.71\%$). See Fig. 1, which also provides an overview of the full pipeline. However, when using classical performance metrics such as FID and CLIP scores the improvements of *SDXL* over previous methods are not reflected as shown in Fig. 13 and discussed in App. E. This aligns with and further backs the findings of Kirstain et al. (2023).

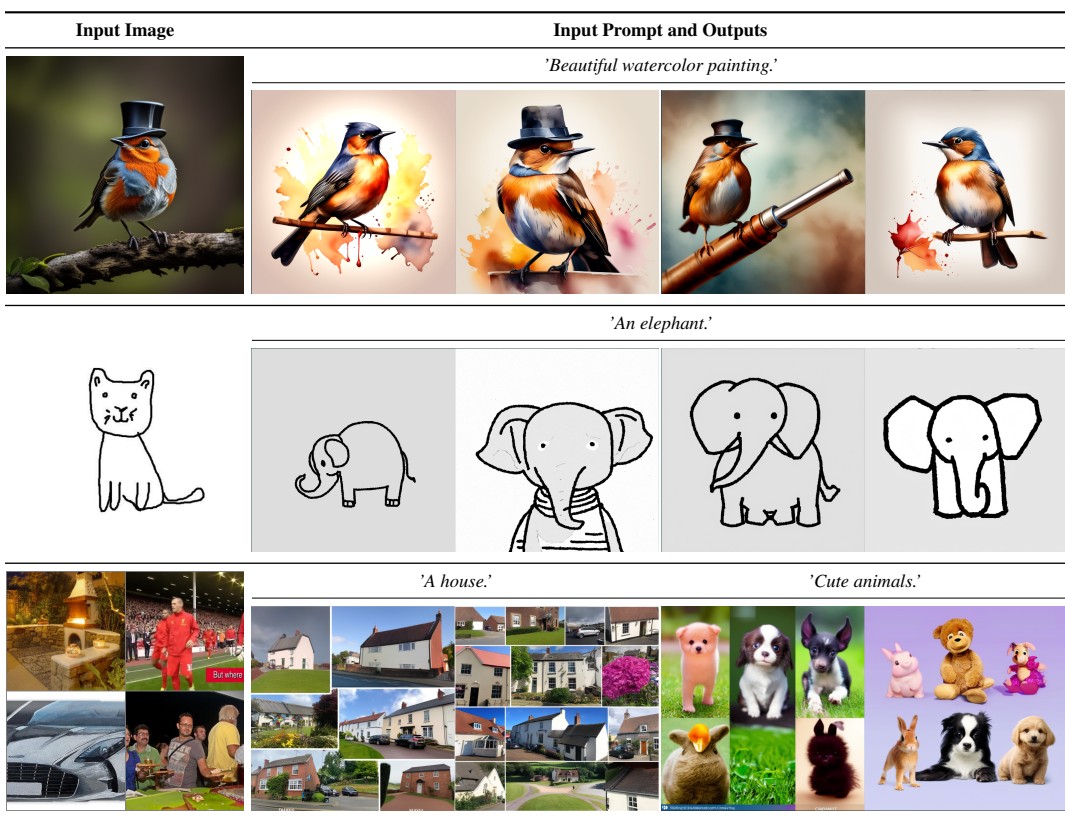

Figure 7: Adding multimodal control: Replacing the pooled text representations of CLIP Radford et al. (2021), which were used during training, with CLIP image features turns *SDXL* into a multimodal image generator for text-controlled image editing, which can even transfer abstract concepts such as "image grid" (bottom row) from the input image to its output. In contrast to previous work Balaji et al. (2022), this does not require joint image-text-conditioned pretraining but only 1000 finetuning steps of a single network layer.

## 2.6    MULTIMODAL CONTROL

Starting with *SDEdit* (Meng et al., 2021), adding image guidance beyond plain text has been a major focus of numerous recent works (Zhang & Agrawala, 2023; Mou et al., 2023; Ruiz et al., 2023; Kawar et al., 2023; Hertz et al., 2022), both with and without further training of the base model. In

this section, we describe a simple and efficient approach to turning *SDXL* into a model guided by *both* text prompts and input images.

As described in Sec. 2.1, we modify the original *Stable Diffusion* architecture by considering not only the text embedding sequence, but also the pooled (global) text representation of the CLIP-G text encoder. By taking advantage of the fact that the pooled CLIP feature space is a (globally) shared image-text feature space, we can replace this global text representation with a global image representation from the CLIP-G image encoder. To account for the slight discrepancy between image and text embeddings, we fine-tune the embedding layer that maps the CLIP embedding to the UNet's timestep embedding space (where they are added), and leave the remaining parameters frozen.

Fig. 7 demonstrates *SDXL*'s multimodal processing capabilities after this fine-tuning, where we prompt the model with both an input image and text input. For example, the model is able to extract the concept "grid" from an input image and transfer it to another output controlled by a text prompt (see bottom row, Fig. 7). We note that a similar approach was implemented in (Balaji et al., 2022), utilizing joint image and video training (from scratch) with high dropout rates on the image conditioning. In contrast, using a trained text-to-image model of *SDXL*, we can replace the pooled CLIP embeddings in a zero-shot manner and achieve high quality by finetuning only the embedding layer for a few thousand steps. Note that we only modify the base model and leave the refiner as is.

## 3 CONCLUSION & FUTURE WORK

This report presents an analysis of improvements to the foundation model *Stable Diffusion* for text-to-image synthesis. While we achieve significant improvements in synthesized image quality, prompt adherence and composition, we believe the model may be improved further in the following aspects:

**Single stage:** Currently, we generate the best samples from *SDXL* with a two-stage approach using our refinement model. This results in having to load two large models into memory, hampering accessibility and sampling speed. Future work should investigate ways to provide a single stage.

**Text synthesis:** While the scale and the larger text encoder (OpenCLIP ViT-bigG (Ilharco et al., 2021)) help to improve the text rendering capabilities over previous versions of *Stable Diffusion*, incorporating byte-level tokenizers (Xue et al., 2022; Liu et al., 2023) or simply scaling the model to larger sizes (Yu et al., 2022; Saharia et al., 2022) should further improve text synthesis.

**Architecture:** During the exploration stage of this work, we briefly experimented with transformer-based architectures such as UViT (Hoogeboom et al., 2023) and DiT (Peebles & Xie, 2022), but found no immediate benefit. We remain, however, optimistic that a careful hyperparameter study will eventually enable scaling to much larger transformer-dominated architectures.

**Distillation:** While our improvements over *Stable Diffusion* are significant, they come at the price of increased inference cost (both in VRAM and sampling speed). Future work will thus focus on decreasing the compute needed for inference, and increased sampling speed, for example through guidance- (Meng et al., 2023), knowledge- (Dockhorn et al., 2023; Kim et al., 2023; Li et al., 2023) and progressive distillation (Salimans & Ho, 2022; Berthelot et al., 2023; Meng et al., 2023).

Finally, our model is trained in the discrete-time formulation of (Ho et al., 2020), and requires *offset-noise* (Guttenberg & CrossLabs, 2023; Lin et al., 2023) for aesthetically pleasing results. The EDM-framework of Karras et al. (2022) is a promising candidate for future model training, as its formulation in continuous time allows for increased sampling flexibility and does not require noise-schedule corrections.

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
