Hugo Touvron, Thibaut Lavril, Gautier Izacard, Xavier Martinet, Marie-Anne Lachaux, Timothée Lacroix, Baptiste Rozière, Naman Goyal, Eric Hambro, Faisal Azhar, Aurelien Rodriguez, Armand Joulin, Edouard Grave, and Guillaume Lample. LLaMA: Open and Efficient Foundation Language Models. *arXiv:2302.13971*, 2023.

Jialiang Wei, Anne-Lise Courbis, Thomas Lambolais, Binbin Xu, Pierre Louis Bernard, and Gérard Dray. Boosting gui prototyping with diffusion models. *arXiv preprint arXiv:2306.06233*, 2023.

Linting Xue, Aditya Barua, Noah Constant, Rami Al-Rfou, Sharan Narang, Mihir Kale, Adam Roberts, and Colin Raffel. Byt5: Towards a token-free future with pre-trained byte-to-byte models, 2022.

Jiahui Yu, Yuanzhong Xu, Jing Yu Koh, Thang Luong, Gunjan Baid, Zirui Wang, Vijay Vasudevan, Alexander Ku, Yinfei Yang, Burcu Karagol Ayan, Ben Hutchinson, Wei Han, Zarana Parekh, Xin Li, Han Zhang, Jason Baldridge, and Yonghui Wu. Scaling autoregressive models for content-rich text-to-image generation, 2022.

Lvmin Zhang and Maneesh Agrawala. Adding conditional control to text-to-image diffusion models. *arXiv:2302.05543*, 2023.

Richard Zhang, Phillip Isola, Alexei A. Efros, Eli Shechtman, and Oliver Wang. The unreasonable effectiveness of deep features as a perceptual metric, 2018.

# Appendix

## A  LIMITATIONS

*A close up of a handpalm with leaves growing from it.*

*An empty fireplace with a television above it. The TV shows a lion hugging a giraffe.*

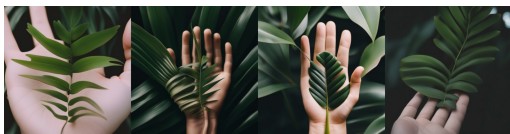 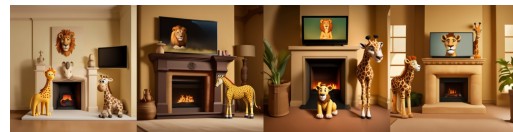

*A grand piano with a white bench.*

*Three quarters view of a rusty old red pickup truck with white doors and a smashed windshield.*

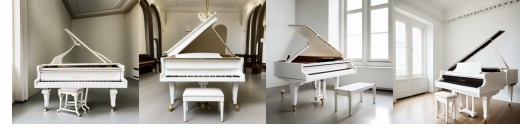 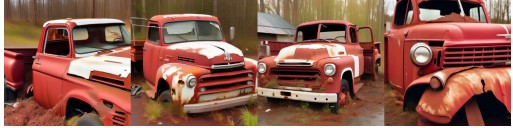

Figure 8: Failure cases of *SDXL* despite large improvements compared to previous versions of *Stable Diffusion*, the model sometimes still struggles with very complex prompts involving detailed spatial arrangements and detailed descriptions (e.g. top left example). Moreover, hands are not yet always correctly generated (e.g. top left) and the model sometimes suffers from two concepts bleeding into one another (e.g. bottom right example). All examples are random samples generated with 50 steps of the DDIM sampler Song et al. (2020a) and cfg-scale 8.0 Ho & Salimans (2022).

While our model has demonstrated impressive capabilities in generating realistic images and synthesizing complex scenes, it is important to acknowledge its inherent limitations. Understanding these limitations is crucial for further improvements and ensuring responsible use of the technology.

Firstly, the model may encounter challenges when synthesizing intricate structures, such as human hands (see Fig. 8, top left). Although it has been trained on a diverse range of data, the complexity of human anatomy poses a difficulty in achieving accurate representations consistently. This limitation suggests the need for further scaling and training techniques specifically targeting the synthesis of fine-grained details. A reason for this occurring might be that hands and similar objects appear with very high variance in photographs and it is hard for the model to extract the knowledge of the real 3D shape and physical limitations in that case.

Secondly, while the model achieves a remarkable level of realism in its generated images, it is important to note that it does not attain perfect photorealism. Certain nuances, such as subtle lighting effects or minute texture variations, may still be absent or less faithfully represented in the generated images. This limitation implies that caution should be exercised when relying solely on model-generated visuals for applications that require a high degree of visual fidelity.

Furthermore, the model's training process heavily relies on large-scale datasets, which can inadvertently introduce social and racial biases. As a result, the model may inadvertently exacerbate these biases when generating images or inferring visual attributes.

In certain cases where samples contain multiple objects or subjects, the model may exhibit a phenomenon known as "concept bleeding". This issue manifests as the unintended merging or overlap of distinct visual elements. For instance, in Fig. 15, an orange sunglass is observed, which indicates an instance of concept bleeding from the orange sweater. Another case of this can be seen in Fig. 9, the penguin is supposed to have a "blue hat" and "red gloves", but is instead generated with blue gloves and a red hat. Recognizing and addressing such occurrences is essential for refining the model's ability to accurately separate and represent individual objects within complex scenes. The root cause of this may lie in the used pretrained text-encoders: firstly, they are trained to compress all information into a single token, so they may fail at binding only the right attributes and objects, Feng et al. (2023) mitigate this issue by explicitly encoding word relationships into the encoding.

Secondly, the contrastive loss may also contribute to this, since negative examples with a different binding are needed within the same batch (Ramesh, 2022).

Additionally, while our model represents a significant advancement over previous iterations of *SD*, it still encounters difficulties when rendering long, legible text. Occasionally, the generated text may contain random characters or exhibit inconsistencies, as illustrated in Fig. 9. Overcoming this limitation requires further investigation and development of techniques that enhance the model's text generation capabilities, particularly for extended textual content — see for example the work of Liu et al. (2023), who propose to enhance text rendering capabilities via character-level text tokenizers. Alternatively, scaling the model does further improve text synthesis (Yu et al., 2022; Saharia et al., 2022).

In conclusion, our model exhibits notable strengths in image synthesis, but it is not exempt from certain limitations. The challenges associated with synthesizing intricate structures, achieving perfect photorealism, further addressing biases, mitigating concept bleeding, and improving text rendering highlight avenues for future research and optimization.

# B  DIFFUSION MODELS

In this section, we give a concise summary of DMs. We consider the continuous-time DM framework (Song et al., 2020b) and follow the presentation of Karras et al. (2022). Let $p_{\text{data}}(\mathbf{x}_0)$ denote the data distribution and let $p(\mathbf{x}; \sigma)$ be the distribution obtained by adding i.i.d. $\sigma^2$-variance Gaussian noise to the data. For sufficiently large $\sigma_{\max}$, $p(\mathbf{x}; \sigma_{\max^2})$ is almost indistinguishable from $\sigma_{\max}^2$-variance Gaussian noise. Capitalizing on this observation, DMs sample high variance Gaussian noise $\mathbf{x}_M \sim \mathcal{N}(\mathbf{0}, \sigma_{\max^2})$ and sequentially denoise $\mathbf{x}_M$ into $\mathbf{x}_i \sim p(\mathbf{x}_i; \sigma_i)$, $i \in \{0, \ldots, M\}$, with $\sigma_i < \sigma_{i+1}$ and $\sigma_M = \sigma_{\max}$. For a well-trained DM and $\sigma_0 = 0$ the resulting $\mathbf{x}_0$ is distributed according to the data.

**Sampling.** In practice, this iterative denoising process explained above can be implemented through the numerical simulation of the *Probability Flow* ordinary differential equation (ODE) (Song et al., 2020b)

$$d\mathbf{x} = -\dot{\sigma}(t)\sigma(t)\nabla_{\mathbf{x}} \log p(\mathbf{x}; \sigma(t)) \, dt, \tag{1}$$

where $\nabla_{\mathbf{x}} \log p(\mathbf{x}; \sigma)$ is the *score function* (Hyvärinen & Dayan, 2005). The schedule $\sigma(t): [0, 1] \to \mathbb{R}_+$ is user-specified and $\dot{\sigma}(t)$ denotes the time derivative of $\sigma(t)$. Alternatively, we may also numerically simulate a stochastic differential equation (SDE) (Song et al., 2020b; Karras et al., 2022):

$$d\mathbf{x} = \underbrace{-\dot{\sigma}(t)\sigma(t)\nabla_{\mathbf{x}} \log p(\mathbf{x}; \sigma(t)) \, dt}_{\text{Probability Flow ODE; see Eq. (1)}} \underbrace{-\beta(t)\sigma^2(t)\nabla_{\mathbf{x}} \log p(\mathbf{x}; \sigma(t)) \, dt + \sqrt{2\beta(t)}\sigma(t) \, d\omega_t}_{\text{Langevin diffusion component}}, \tag{2}$$

where $d\omega_t$ is the standard Wiener process. In principle, simulating either the Probability Flow ODE or the SDE above results in samples from the same distribution.

**Training.** DM training reduces to learning a model $s_{\boldsymbol{\theta}}(\mathbf{x}; \sigma)$ for the score function $\nabla_{\mathbf{x}} \log p(\mathbf{x}; \sigma)$. The model can, for example, be parameterized as $\nabla_{\mathbf{x}} \log p(\mathbf{x}; \sigma) \approx s_{\boldsymbol{\theta}}(\mathbf{x}; \sigma) = (D_{\boldsymbol{\theta}}(\mathbf{x}; \sigma) - \mathbf{x})/\sigma^2$ (Karras et al., 2022), where $D_{\boldsymbol{\theta}}$ is a learnable *denoiser* that, given a noisy data point $\mathbf{x}_0 + \mathbf{n}$, $\mathbf{x}_0 \sim p_{\text{data}}(\mathbf{x}_0)$, $\mathbf{n} \sim \mathcal{N}(\mathbf{0}, \sigma^2 \boldsymbol{I}_d)$, and conditioned on the noise level $\sigma$, tries to predict the clean $\mathbf{x}_0$. The denoiser $D_{\boldsymbol{\theta}}$ (or equivalently the score model) can be trained via *denoising score matching* (DSM)

$$\mathbb{E}_{(\mathbf{x}_0, \mathbf{c}) \sim p_{\text{data}}(\mathbf{x}_0, \mathbf{c}), (\sigma, \mathbf{n}) \sim p(\sigma, \mathbf{n})} \left[ \lambda_\sigma \| D_{\boldsymbol{\theta}}(\mathbf{x}_0 + \mathbf{n}; \sigma, \mathbf{c}) - \mathbf{x}_0 \|_2^2 \right], \tag{3}$$

where $p(\sigma, \mathbf{n}) = p(\sigma)\mathcal{N}(\mathbf{n}; \mathbf{0}, \sigma^2)$, $p(\sigma)$ is a distribution over noise levels $\sigma$, $\lambda_\sigma: \mathbb{R}_+ \to \mathbb{R}_+$ is a weighting function, and $\mathbf{c}$ is an arbitrary conditioning signal, e.g., a class label, a text prompt, or a combination thereof. In this work, we choose $p(\sigma)$ to be a discrete distributions over 1000 noise levels and set $\lambda_\sigma = \sigma^{-2}$ similar to prior works (Ho et al., 2020; Rombach et al., 2021; Sohl-Dickstein et al., 2015).

**Classifier-free guidance.** Classifier-free guidance (Ho & Salimans, 2022) is a technique to guide the iterative sampling process of a DM towards a conditioning signal $\mathbf{c}$ by mixing the predictions of a conditional and an unconditional model

$$D^w(\mathbf{x}; \sigma, \mathbf{c}) = (1 + w)D(\mathbf{x}; \sigma, \mathbf{c}) - wD(\mathbf{x}; \sigma), \tag{4}$$

where $w \geq 0$ is the *guidance strength*. In practice, the unconditional model can be trained jointly alongside the conditional model in a single network by randomly replacing the conditioning signal $\mathbf{c}$ with a null embedding in Eq. (3), e.g., 10% of the time (Ho & Salimans, 2022). Classifier-free guidance is widely used to improve the sampling quality, trading for diversity, of text-to-image DMs (Nichol et al., 2021; Rombach et al., 2021).

## C  COMPARISON TO THE STATE OF THE ART

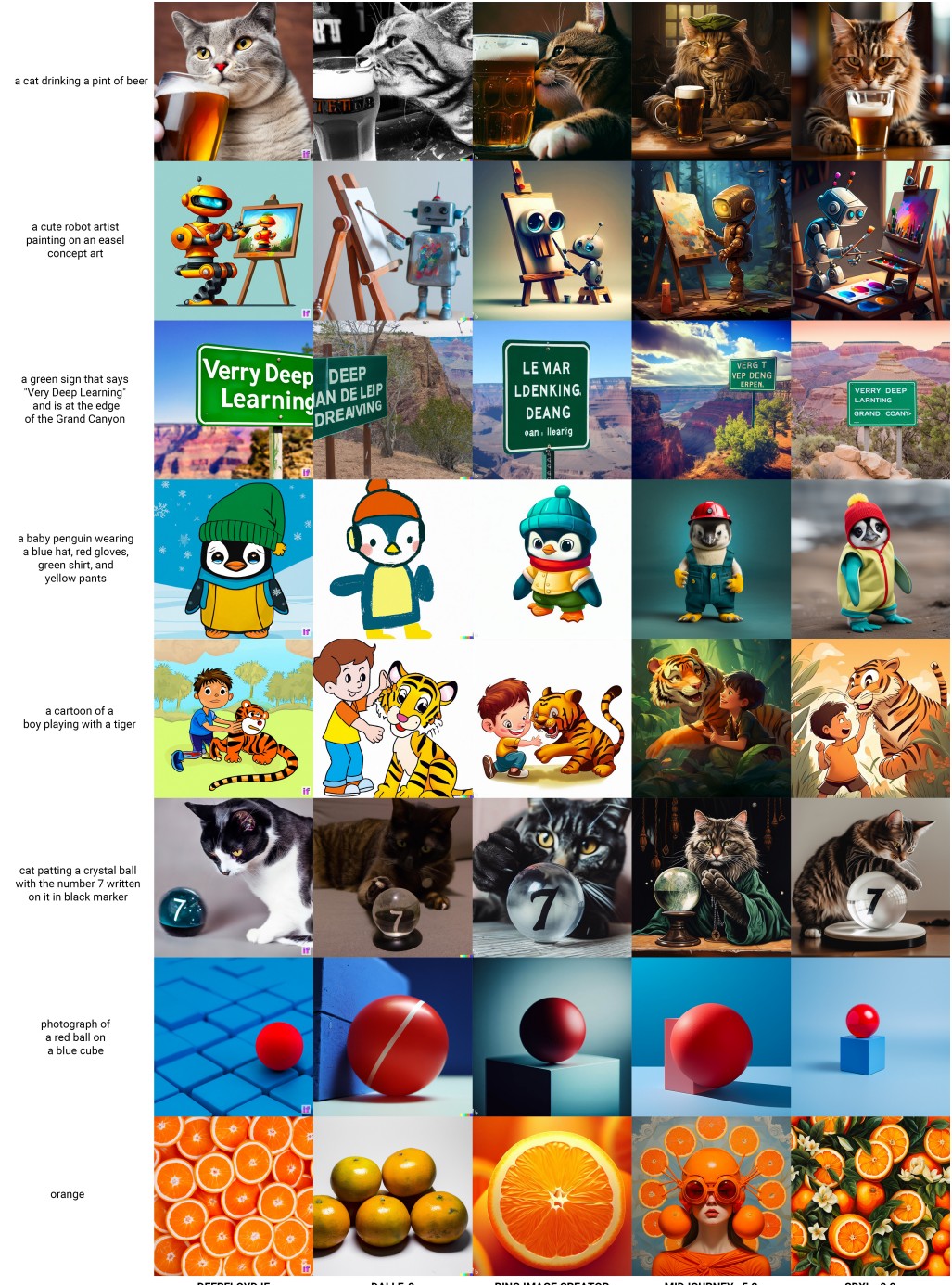

Figure 9: Qualitative comparison of *SDXL* with DeepFloyd IF, DALLE-2, Bing Image Creator, and Midjourney v5.2. To mitigate any bias arising from cherry-picking, Parti (P2) prompts were randomly selected. Seed 3 was uniformly applied across all models in which such a parameter could be designated. For models without a seed-setting feature, the first generated image is included.

## D COMPARISON TO MIDJOURNEY V5.1

### D.1 OVERALL VOTES

To asses the generation quality of *SDXL* we perform a user study against the state of the art text-to-image generation platform Midjourney[1]. As the source for image captions we use the PartiPrompts (P2) benchmark (Yu et al., 2022), that was introduced to compare large text-to-image model on various challenging prompts.

For our study, we choose five random prompts from each category, and generate four $1024 \times 1024$ images by both Midjourney (v5.1, with a set seed of 2) and *SDXL* for each prompt. These images were then presented to the AWS GroundTruth taskforce, who voted based on adherence to the prompt. The results of these votes are illustrated in Fig. 10. Overall, there is a slight preference for *SDXL* over Midjourney in terms of prompt adherence.

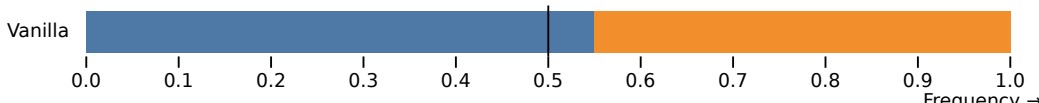

Figure 10: Results from 17,153 user preference comparisons between *SDXL* v0.9 and Midjourney v5.1, which was the latest version available at the time. The comparisons span all "categories" and "challenges" in the PartiPrompts (P2) benchmark. Notably, *SDXL* was favored 54.9% of the time over Midjourney V5.1. Preliminary testing indicates that the recently-released Midjourney V5.2 has lower prompt comprehension than its predecessor, but the laborious process of generating multiple prompts hampers the speed of conducting broader tests.

### D.2 CATEGORY & CHALLENGE COMPARISONS ON PARTIPROMPTS (P2)

Each prompt from the P2 benchmark is organized into a category and a challenge, each focus on different difficult aspects of the generation process. We show the comparisons for each category (Fig. 11) and challenge (Fig. 12) of P2 below. In four out of six categories *SDXL* outperforms Midjourney, and in seven out of ten challenges there is no significant difference between both models or *SDXL* outperforms Midjourney.

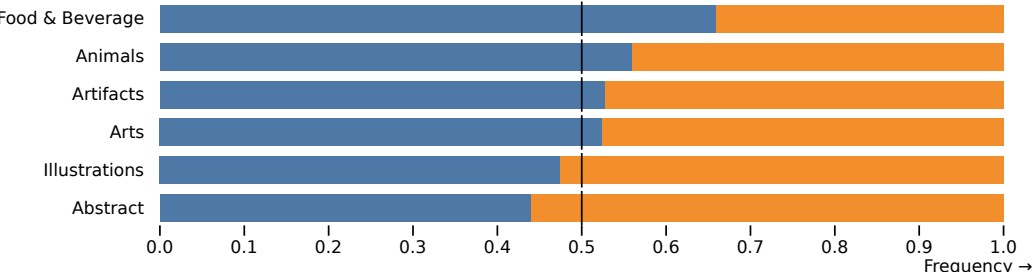

Figure 11: User preference comparison of *SDXL* (without refinement model) and Midjourney V5.1 across particular text categories. *SDXL* outperforms Midjourney V5.1 in all but two categories.

---

[1]We compare against v5.1 since that was the best version available at that time.

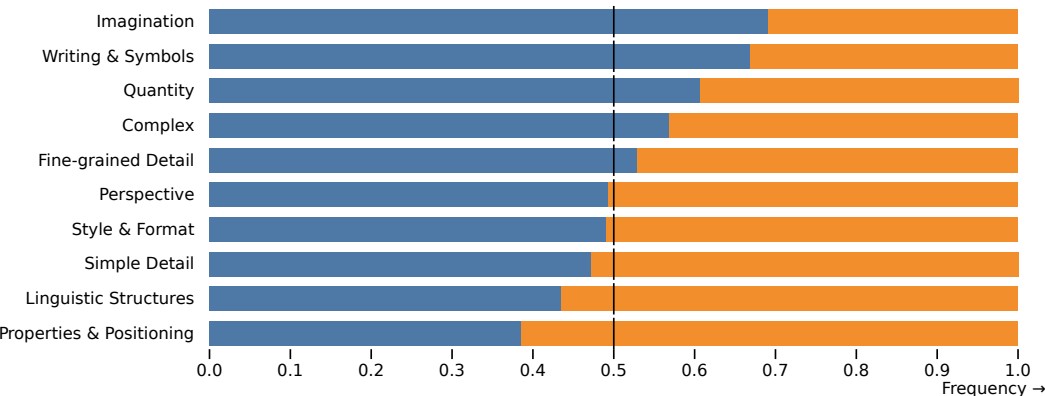

Figure 12: Preference comparisons of *SDXL* (with refinement model) to Midjourney V5.1 on complex prompts. *SDXL* either outperforms or is statistically equal to Midjourney V5.1 in 7 out of 10 categories.

# E  ON FID ASSESSMENT OF GENERATIVE TEXT-IMAGE FOUNDATION MODELS

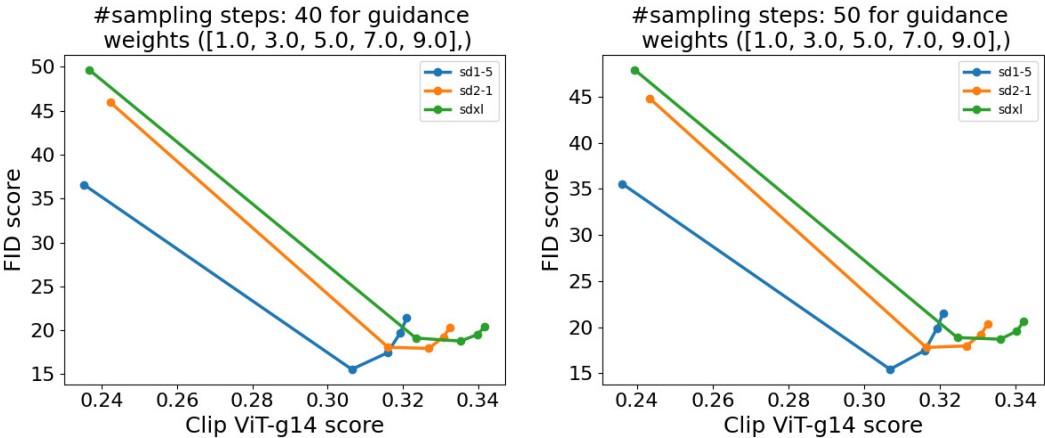

Figure 13:  Plotting FID vs CLIP score for different cfg scales. *SDXL* shows only slightly improved text-alignment, as measured by CLIP-score, compared to previous versions that do not align with the judgement of human evaluators. Even further and similar as in Kirstain et al. (2023), FID are worse than for both *SD-1.5* and *SD-2.1*, while human evaluators clearly prefer the generations of *SD-XL* over those of these previous models.

Throughout the last years it has been common practice for generative text-to-image models to assess FID- (Heusel et al., 2017) and CLIP-scores (Radford et al., 2021; Ramesh et al., 2021) in a zero-shot setting on complex, small-scale text-image datasets of natural images such as COCO (Lin et al., 2015). However, with the advent of foundational text-to-image models (Saharia et al., 2022; Ramesh et al., 2022; Rombach et al., 2021; Balaji et al., 2022), which are not only targeting visual compositionality, but also at other difficult tasks such as deep text understanding, fine-grained distinction between unique artistic styles and especially a pronounced sense of visual aesthetics, this particular form of model evaluation has become more and more questionable. Kirstain et al. (2023) demonstrates that COCO zero-shot FID is *negatively correlated* with visual aesthetics, and such measuring the generative performance of such models should be rather done by human evaluators. We investigate this for *SDXL* and visualize FID-vs-CLIP curves in Fig. 13 for 10k text-image pairs from COCO (Lin et al., 2015). Despite its drastically improved performance as measured quantitatively by asking human assessors (see Fig. 1) as well as qualitatively (see Fig. 4 and Fig. 15), *SDXL* does *not* achieve better FID scores than the previous *SD* versions. Contrarily, FID for *SDXL* is the worst of all three compared models while only showing slightly improved CLIP-scores (measured with OpenClip ViT g-14). Thus, our results back the findings of Kirstain et al. (2023) and further emphasize the need

for additional quantitative performance scores, specifically for text-to-image foundation models. All scores have been evaluated based on 10k generated examples.

# F ADDITIONAL COMPARISON BETWEEN SINGLE- AND TWO-STAGE *SDXL* PIPELINE

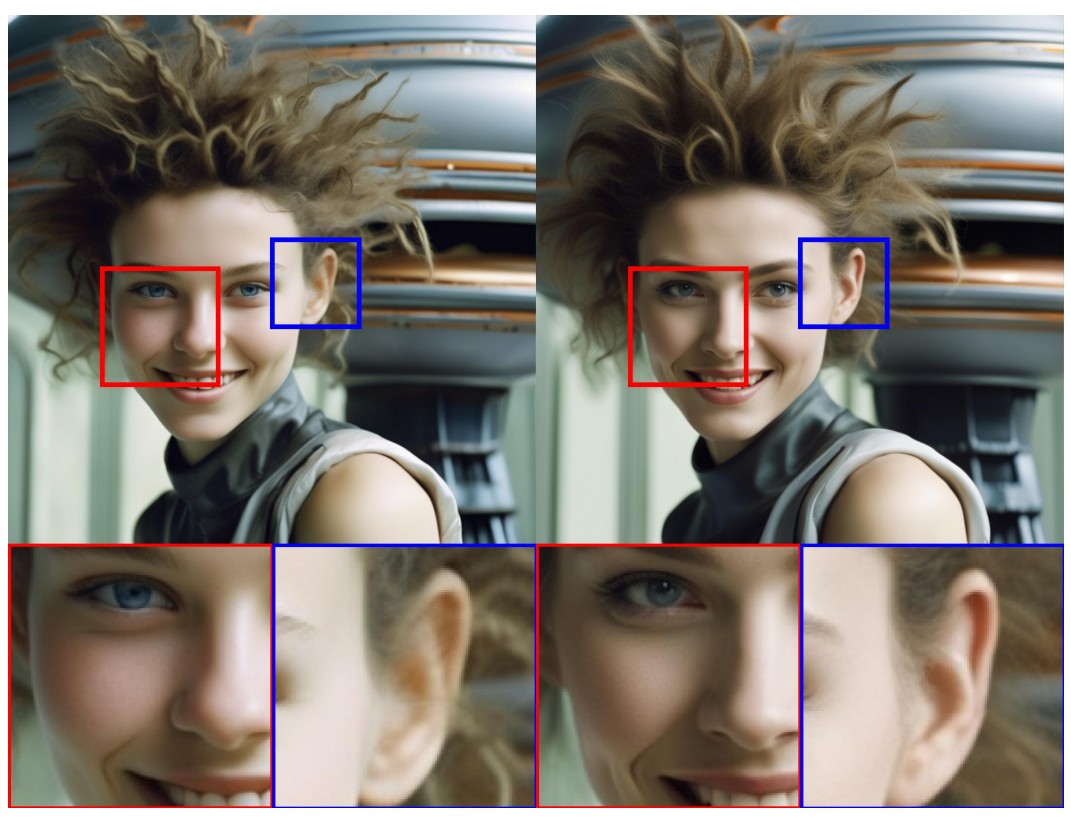

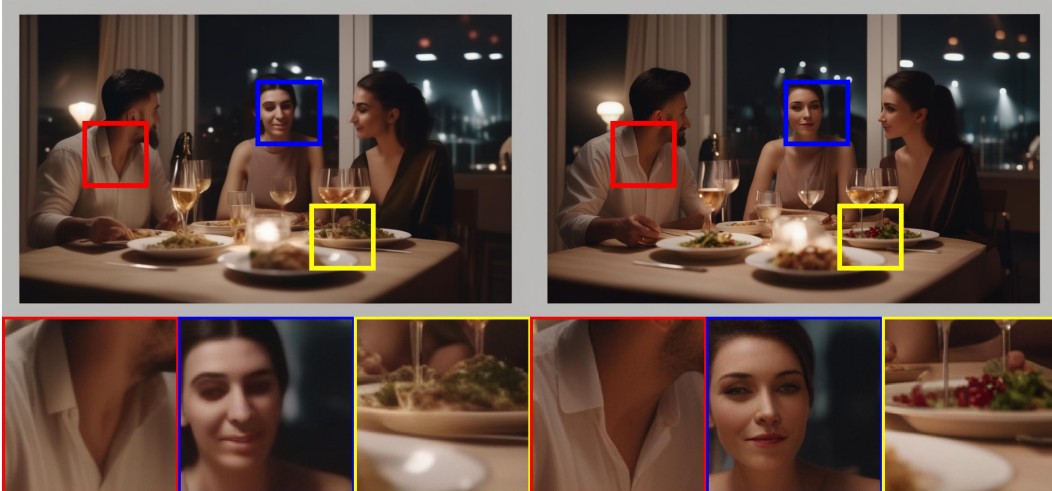

Figure 14: *SDXL* samples (with zoom-ins) without (left) and with (right) the refinement model discussed. Prompt: (*top*) "close up headshot, futuristic young woman, wild hair sly smile in front of gigantic UFO, dslr, sharp focus, dynamic composition" (*bottom*) "Three people having dinner at a table at new years eve, cinematic shot, 8k". Zoom-in for details.

## G   COMPARISON BETWEEN *SD 1.5* VS. *SD 2.1* VS. *SDXL*

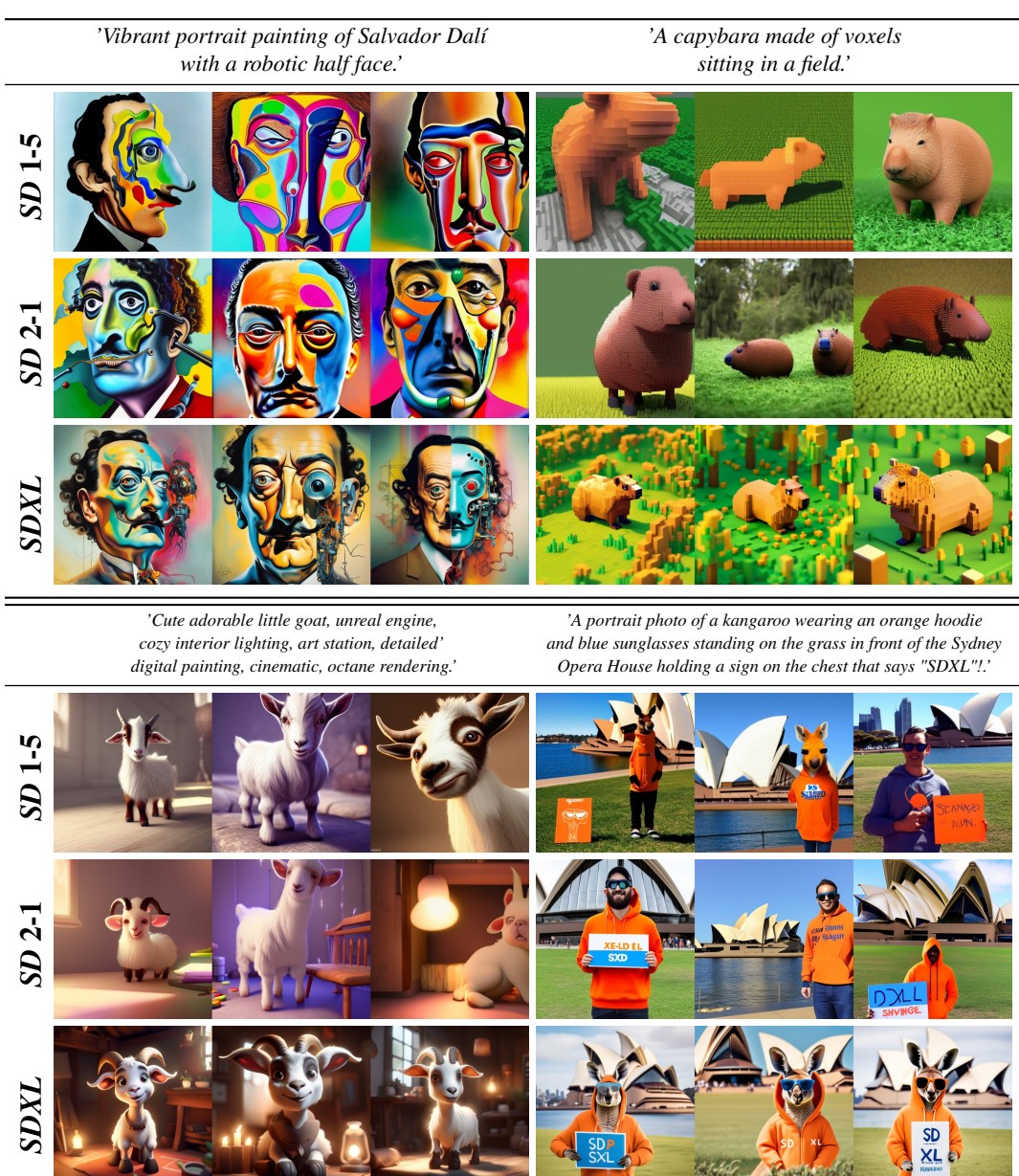

Figure 15:   Additional results for the comparison of the output of *SDXL* with previous versions of *Stable Diffusion*. For each prompt, we show 3 random samples of the respective model for 50 steps of the DDIM sampler Song et al. (2020a) and cfg-scale 8.0 Ho & Salimans (2022)

| *'Monster Baba yaga house with in a forest, dark horror style, black and white.'* | *'A young badger delicately sniffing a yellow rose, richly textured oil painting.'* |
|---|---|

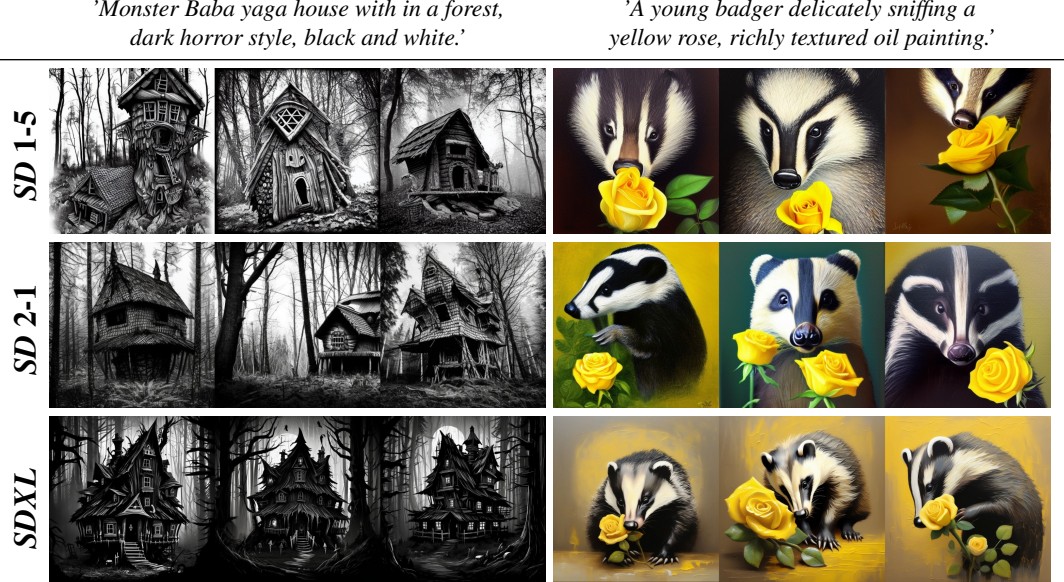

Figure 16: Additional results for the comparison of the output of *SDXL* with previous versions of *Stable Diffusion*. For each prompt, we show 3 random samples of the respective model for 50 steps of the DDIM sampler Song et al. (2020a) and cfg-scale 8.0 Ho & Salimans (2022).

## H  MULTI-ASPECT TRAINING HYPERPARAMETERS

We use the following image resolutions for mixed-aspect ratio finetuning as described in Sec. 2.3.

| Height | Width | Aspect Ratio | Height | Width | Aspect Ratio |
|---|---|---|---|---|---|
| 512 | 2048 | 0.25 | 1024 | 1024 | 1.0 |
| 512 | 1984 | 0.26 | 1024 | 960 | 1.07 |
| 512 | 1920 | 0.27 | 1088 | 960 | 1.13 |
| 512 | 1856 | 0.28 | 1088 | 896 | 1.21 |
| 576 | 1792 | 0.32 | 1152 | 896 | 1.29 |
| 576 | 1728 | 0.33 | 1152 | 832 | 1.38 |
| 576 | 1664 | 0.35 | 1216 | 832 | 1.46 |
| 640 | 1600 | 0.4 | 1280 | 768 | 1.67 |
| 640 | 1536 | 0.42 | 1344 | 768 | 1.75 |
| 704 | 1472 | 0.48 | 1408 | 704 | 2.0 |
| 704 | 1408 | 0.5 | 1472 | 704 | 2.09 |
| 704 | 1344 | 0.52 | 1536 | 640 | 2.4 |
| 768 | 1344 | 0.57 | 1600 | 640 | 2.5 |
| 768 | 1280 | 0.6 | 1664 | 576 | 2.89 |
| 832 | 1216 | 0.68 | 1728 | 576 | 3.0 |
| 832 | 1152 | 0.72 | 1792 | 576 | 3.11 |
| 896 | 1152 | 0.78 | 1856 | 512 | 3.62 |
| 896 | 1088 | 0.82 | 1920 | 512 | 3.75 |
| 960 | 1088 | 0.88 | 1984 | 512 | 3.88 |
| 960 | 1024 | 0.94 | 2048 | 512 | 4.0 |

# I  PSEUDO-CODE FOR CONDITIONING CONCATENATION ALONG THE CHANNEL AXIS

```python
from einops import rearrange
import torch

batch_size=16
# channel dimension of pooled output of text encoder(s)
pooled_dim = 512

def fourier_embedding(inputs, outdim=256, max_period=10000):
    """
    Classical sinusoidal timestep embedding
    as commonly used in diffusion models
    :param inputs: batch of integer scalars shape [b,]
    :param outdim: embedding dimension
    :param max_period: max freq added
    :return: batch of embeddings of shape [b, outdim]
    """
    ...

def cat_along_channel_dim(
        x:torch.Tensor,) -> torch.Tensor:
    if x.ndim == 1:
        x = x[...,None]
    assert x.ndim == 2
    b, d_in = x.shape
    x = rearrange(x, "b din -> (b din)")
    # fourier fn adds additional dimension
    emb = fourier_embedding(x)
    d_f = emb.shape[-1]
    emb = rearrange(emb, "(b din) df -> b (din df)",
                    b=b, din=d_in, df=d_f)
    return emb

def concat_embeddings(
        # batch of size and crop conditioning cf. Sec. 3.2
        c_size:torch.Tensor,
        c_crop:torch.Tensor,
        # batch of aspect ratio conditioning cf. Sec. 3.3
        c_ar:torch.Tensor,
        # final output of text encoders after pooling cf. Sec. 3.1
        c_pooled_txt:torch.Tensor, ) -> torch.Tensor:
    # fourier feature for size conditioning
    c_size_emb = cat_along_channel_dim(c_size)
    # fourier feature for size conditioning
    c_crop_emb = cat_along_channel_dim(c_crop)
    # fourier feature for size conditioning
    c_ar_emb = cat_along_channel_dim(c_ar)
    # the concatenated output is mapped to the same
    # channel dimension than the noise level conditioning
    # and added to that conditioning before being fed to the unet
    return torch.cat([c_pooled_txt,
                      c_size_emb,
                      c_crop_emb,
                      c_ar_emb], dim=1)

# simulating c_size and c_crop as in Sec. 3.2
c_size=torch.zeros((batch_size, 2)).long()
c_crop=torch.zeros((batch_size, 2)).long()
# simulating c_ar and pooled text encoder output as in Sec. 3.3
c_ar=torch.zeros((batch_size, 2)).long()
c_pooled=torch.zeros((batch_size, pooled_dim)).long()

# get concatenated embedding
c_concat = concat_embeddings(c_size, c_crop, c_ar, c_pooled)
```

Figure 17: Python code for concatenating the additional conditionings introduced in Secs. 2.1 to 2.3 along the channel dimension.

| Config | Batch Size | EMA | Global Steps | rFID [↓] | PSNR [↑] | SSIM [↑] | LPIPS [↓] |
|--------|-----------|-----|--------------|----------|----------|----------|-----------|
| B      | 256       | ✗   | 500k         | 7.52     | 24.30    | 0.71     | 1.27      |
| Be     | 256       | ✓   | 500k         | 7.35     | 24.33    | 0.72     | 1.24      |
| A1     | 8         | ✗   | 500k         | 9.14     | 24.42    | 0.72     | 1.32      |
| A1e    | 8         | ✓   | 500k         | 8.85     | 24.51    | 0.72     | 1.29      |
| A2     | 8         | ✗   | 1.9M         | 6.28     | 25.05    | 0.74     | 1.08      |
| A2e    | 8         | ✓   | 1.9M         | 5.57     | 25.05    | 0.74     | 1.07      |

Table 4: Evaluting two AE models trained with different batch sizes, see App. J for details.

## J  AUTOENCODER TRAININGS-HYPERPARAMETER

To assess the effect of (1) larger batch size and (2) using an exponential moving average (EMA) of the weights discussed in Sec. 2.4, we conduct two experiments, where we train the autoencoder (from scratch) on with (a) a total batch size of 8 and (b) a total batch size of 256. These models are trained as an exploratory experiment and not trained until convergence. For both experiments, we evaluate the reconstruction performance on a fixed 10k subset of the COCO2014 validation set, comparing EMA and non-EMA weights for both models. We report rFID, PSNR, SSIM, and LPIPS in a Tab. 4. Example reconstructions can be seen in Fig. 18.

We see, that:

1. In all our evaluations the EMA variants give a consistent improvement compared to the non-EMA variant. This effect is especially amplified with more update steps (compare A1(e) and A2(e) in Tab. 4).

2. To evaluate the effect of the large batch size we first compare the models when they are both trained for the same number of gradient updates (A1(e) and B(e)): In this case, the large batch size provides a clear benefit in terms of rFID: 7.34 vs 8.85, while the other metrics are relatively close. However, when we evaluate the models after they have trained for roughly the same wall time (500k (B) and 1.9M (A2) update steps), the larger number of updates is a more important factor, with A2e outperforming Be in all metrics.
   We believe that training until convergence might change this outcome, since we can see from the B/A1 evaluation that the large batch size provide a better update direction for the weights. This will likely help in finding a loss-minimum.

From this experiment, we conclude that EMA-tracking has a significant effect on our performance, but we believe that the choice of the trade-off between batch size and update steps requires more thorough investigation in a future work.

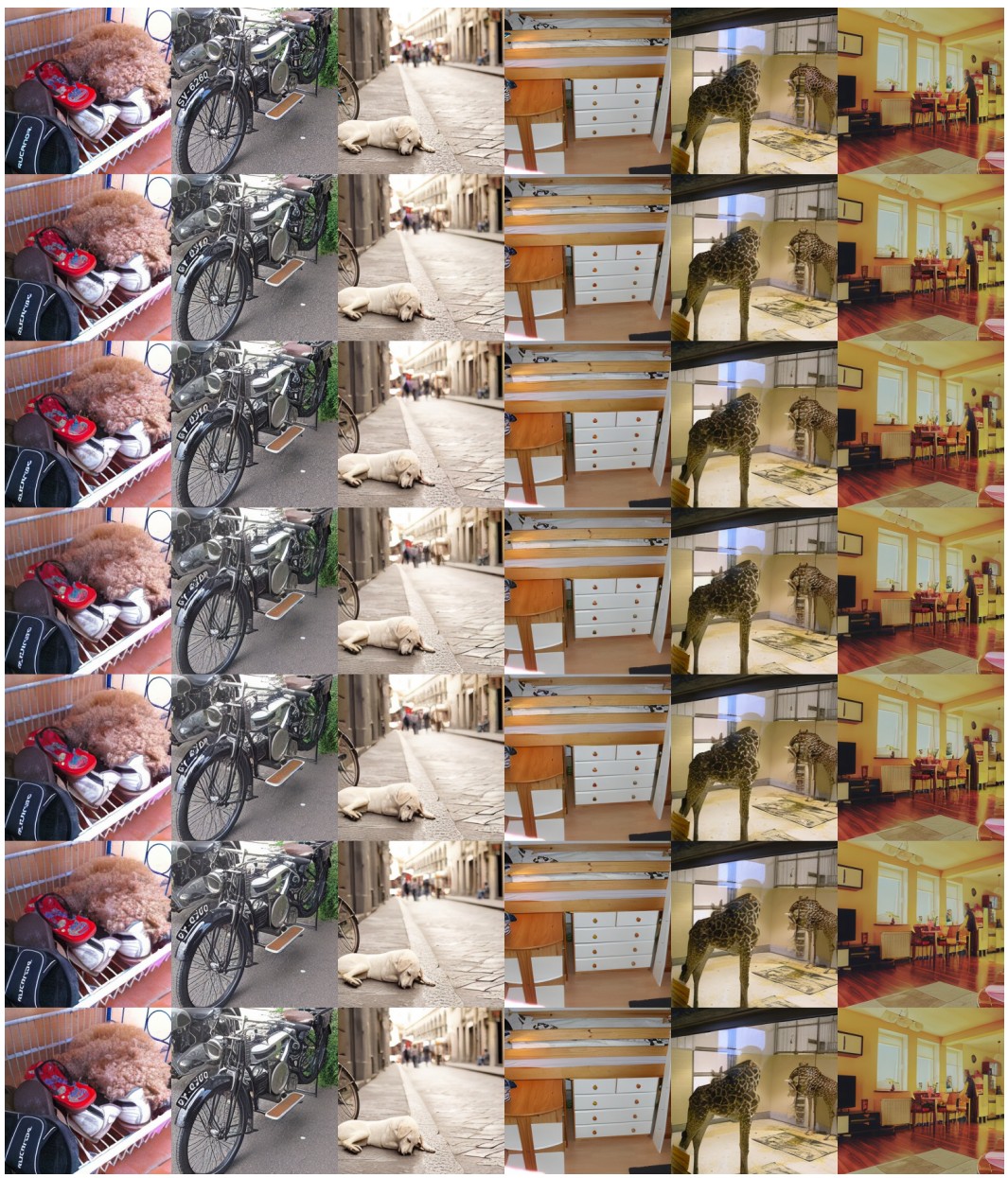

Figure 18: Qualitative evaluation of the AEs trained for App. J. The rows correspond to: real images, B, Be, A1, A1e, A2, A2e (cf. Tab. 4).