# OpenReview forum: "SDXL: Improving Latent Diffusion Models for High-Resolution Image Synthesis"
_ICLR.cc/2024/Conference — ICLR 2024 spotlight_

### Official Review · Reviewer_rcq7 · 2023-10-30

**Soundness:** 4 excellent
**Presentation:** 4 excellent
**Contribution:** 4 excellent
**Rating:** 8
**Confidence:** 4

**Summary:**

This paper introduces the following techniques to improve Stable Diffusion and it achieves much better performance according to user study. The trained model is also promised to be released publicly:
1. A more scalable model architecture: modifications on transformer blocks, more powerful text encoders.
2. Enable conditioning on image size and cropping.
3. Training with multiple aspect ratio.
4. Better Autoencoder.
5. Additional refinement stage enables more high-resolution details.
6. Additional finetuning that conditions on pooled text embedding enables multi-modal control during inference.

**Strengths:**

1. The paper is well-written with clear high-level intuitions/ideas, low-level implementation details, as well as visualizations that justify the improvements. It is easy to follow and understand even for the audience that only have very basic knowledge or practice about diffusion models.

2. The problems the authors are trying to tackle are very realistic in practice and the their proposed solutions are simple and effective:
- in order to fully utilize all the training data with varying image resolutions, they propose to add conditioning on image size during training.
- in order to solve the commonly-seen cut-out artifacts during generation (as in Fig. 4), they propose to add conditioning on cropping parameters during training.
- in order to improve high-resolution details, they propose to train another refinement model and use it in a SDEdit way during inference.
- in order to enable multi-modal control during inference, they propose to finetune only the embedding layer of a text-to-image SDXL such that it can condition on pooled CLIP text embedding as well, which could be naively replaced with image CLIP embedding during inference.

3. The authors promised to open-source the model (last paragraph of Sec. 1), plus all the implementation details in both the main text and supplementary materials including the pseudo code in Fig. 17. I believe this work could be a huge add-on to the image generation community and enable more future works that build upon it, just like Stable Diffusion.

**Weaknesses:**

1. Multiple techniques are proposed in this work, therefore a more detailed ablation study would help the audience better understand what’s the influence of each individual component, especially for the two additional conditioning (size and crop).
2. Particularly, I’m very curious on which method would be more critical to address the aspect ratio issue, the proposed cropping conditioning or data bucketing from NovelAI? According to the description in Sec. 2.3, these two techniques are combined together and there’s no separate evaluation on each of them.
3. The rightmost examples in Fig. 5 look confusing to me: does (512, 512) mean the whole images are cropped out and there should be nothing inside (the caption says images are from a 512^2 model)? In this case, why does the top example still show basically the same image as the leftmost one while the bottom example shows mainly the background?
4. The training data of SDXL is not mentioned in the paper. Is it the same as Stable Diffusion, i.e., LAION? If not, would the comparison still be able to justify the effectiveness of proposed techniques since the training data is different?

**Questions:**

It would be great if the authors could respond to the weakness points mentioned above. Thanks!

---

> ### Author Response · Authors · 2023-11-23
>
> Dear Reviewer rcq7,
>
> Thank you for your careful review of our manuscript and your appreciation for our efforts. To address your questions:
>
> In general, crop-conditioning can be applied as a finetuning technique and address issues observed in previous versions of SD when random cropping expansion from training carried over into the synthesized samples. However, as described in Section 2.3, crop-conditioning is particularly beneficial when we train at a fixed resolution, e.g. 512x512 pixels. For the final multi-aspect model, crop-conditioning does not need to be applied. The effectiveness of size conditioning is shown in Tab. 2. In Fig 5 the model is sampled with different inputs for the crop conditoning, but no actual cropping is happening. The rightmost examples in Fig. 5 are OOD, and the behavior here is therefore less predictable.

---

### Official Review · Reviewer_gmj5 · 2023-10-31

**Soundness:** 3 good
**Presentation:** 3 good
**Contribution:** 3 good
**Rating:** 8
**Confidence:** 3

**Summary:**

This method proposes SDXL,  a latent diffusion model for text-to-image synthesis. Compared to previous versions of Stable Diffusion, SDXL leverages a three times larger UNet backbone, achieved by significantly increasing the number of attention blocks and including a second text encoder. Also, the authors propose multiple novel conditioning schemes and train SDXL on multiple aspect ratios. Competitive performance achieved with SoTA models such as Midjourney.

**Strengths:**

1. Very sound elaboration and experiments. The proposed improvement is reasonable and effective.
2. The method is open sourced.

**Weaknesses:**

1. Lacks some comparison with SoTA pixel-space diffusion model such as Simple Diffusion from Google.
2.  Also, SDXL still relied on two-stage training where a high-quality encoder is required. I wonder whether the two stage can be combined and lead to further performance boost?

**Questions:**

1. As discussed in your future work section, can SD be combined into single-stage training and further boost the performance?
2. Can similar training pipeline introduced in this paper be applied to 3D diffusion training, and what are the challenges?

**Details Of Ethics Concerns:**

No.

---

> ### Author Response · Authors · 2023-11-23
>
> Dear Reviewer gmj5,
> Thank you for your careful review of our manuscript and your appreciation of our efforts.
>
> In response to your questions:
>
> - Single-Stage Training: We agree that single-stage training is an exciting avenue for future research, and both [1] and [2] present promising results in this direction. Currently, two-stage, latent diffusion approaches give the best results [5] for text-to-image synthesis and we follow this paradigm.
> - For 3D, the main issue with directly scaling diffusion-based approaches is data scarcity [3]. However, we hypothesize that multimodal approaches that combine explicit 3D training with 2D and video training can eventually help to overcome this issue. The micro-conditioning techniques that we introduced in our paper can be adapted and applied to other modalities, see e.g. [4]
>
> References:
>
> [1]: Emiel Hoogeboom and Jonathan Heek and Tim Salimans: *simple diffusion: End-to-end diffusion for high resolution images*
>
> [2]: Gu et al: *Matryoshka Diffusion Models*
>
> [3]: Deitke el al: *Objaverse-XL: A Universe of 10M+ 3D Objects*
>
> [4]: Blattmann et al: *Stable Video Diffusion: Scaling Latent Video Diffusion Models to Large Datasets*
>
> [5]: Betker et al: *Improving Image Generation with Better Captions*

---

### Official Review · Reviewer_n7kc · 2023-11-01

**Soundness:** 4 excellent
**Presentation:** 4 excellent
**Contribution:** 4 excellent
**Rating:** 8
**Confidence:** 4

**Summary:**

This paper presents SDXL, an extended version of the existing StableDiffusion model, showcasing its training and capabilities. The authors detail the methodology used in image training, including the resolution, crop, top, and left condition techniques, as well as the innovative multi-aspect resolution training approach. Furthermore, the paper sheds light on the advancements in AutoEncoder technology and the implementation of a Refinement model. The overall generation process is structured in two stages, allowing for high quality images. Also authors introduce diverse multimodal controls. This comprehensive explanation highlights the significant enhancements and versatility that SDXL brings to image generation, setting a new standard in the field.

**Strengths:**

This paper not only introduces a variety of innovative training methods but also provides clear intuition and thorough analysis to support them. The novel conditioning techniques and multi-aspect resolution training approaches presented in the manuscript are both groundbreaking and well-substantiated with comprehensive evaluations. The authors have done an excellent job of providing numerical metrics and a wide array of visual comparisons to showcase the effectiveness of their proposed methods. The manuscript is well-written, with a coherent structure that facilitates easy understanding, making it a significant contribution to the field.

**Weaknesses:**

The manuscript appears to lack detailed explanations on how the autoencoder (AE) has been improved or advanced. There is also no clear description of the user study conducted, raising questions about its methodology and implementation. Is it conducted in a manner similar to what is described in the supplementary materials? Providing more information and context on these aspects would greatly enhance the comprehensibility and robustness of the paper.

**Questions:**

- Refinement training : Could you kindly explain about details of refinement training? This training is only for  200 (discrete) noise scales which is [200, 0] ?

- Could you give me some intuition and the effect of using a combination of language models (CLIP ViT-L & OpenCLIP ViT-bigG)? Is it crucial to use all of them?

- I wonder the author's opinion about using TransFormer architecture to train the DMs. Could you share more information on "no immediate benefit"?

**Details Of Ethics Concerns:**

No concerns

---

> ### Author Response · Authors · 2023-11-23
>
> Dear Reviewer n7kc
>
> We appreciate your careful review of our manuscript and your recognition of our efforts.
>
> As we describe in Sec 2.4, the autoencoder has been improved by adding EMA-based weight tracking and increasing the batch size used during training. Since reviewer pUdr also asked about the autoencoder, we answer that in our general response. TL;DR: EMA consistently gives better reconstruction metrics. Larger batch sizes provide better gradient updates.
>
> In response to your further questions:
>
> 1. Yes.
> 2. We empirically found that these CLIP text encoders contain complementary semantic knowledge, for example about visual styles.
> 3. We briefly experimented with the DiT[1] and UViT[2] architectures, but found that the images produced were not of higher quality. However, this could be due to the suboptimal choice of hyperparameters during the short experimentation phase. we remain optimistic that these architectures are good candidates for scaling diffusion models to much larger sizes, and that the simplified architectures may ease engineering efforts.
>
> References:
>
> [1]: William Peebles and Saining Xie: *Scalable Diffusion Models with Transformers*
>
> [2]: Emiel Hoogeboom and Jonathan Heek and Tim Salimans: *simple diffusion: End-to-end diffusion for high resolution images*

---

### Official Review · Reviewer_pUdr · 2023-11-04

**Soundness:** 3 good
**Presentation:** 3 good
**Contribution:** 4 excellent
**Rating:** 8
**Confidence:** 5

**Summary:**

This paper presents SDXL, a latent diffusion model for text-to-image synthesis. SDXL uses a larger U-Net compared to previous Stable Diffusion models, and adds a refiner module to improve visual quality of image samples. During SDXL training, the U-Net is conditioned on image size, image cropping information, and receives training data in multiple aspect ratios. A new VAE is trained, with improved reconstruction performance compared to earlier SD versions.

**Strengths:**

1. SDXL demonstrates impressive text-to-image generation results. It can serve as a strong base model for a broad range of research and applications in downstream image synthesis tasks.
2. Some training and architectural choices are backed with convincing ablation experiments (e.g. Table 2). This can offer some valuable insights for future training of image and video generation models.
3. The refinement model could be an additional contribution, to refine local details of real / generated images.

**Weaknesses:**

1. For "conditioning on cropping" and "multi-aspect training", this paper lacks adequate quantitative experiments to demonstrate their effectiveness.
2. Minor mistakes:
(1) In Figure 3, the image size should be (256, 256) instead of (256, 236).
(2) In Figure 3 and Figure 4, the quotation mark for text prompt is incorrect.

**Questions:**

1. There could more explanations on why your current autoencoder provides better reconstruction performance: which factor contributes more, the EMA choice, or larger batch size?
2. How well does the refinement model work when refining other real images, or images generated by previous SD versions? That is, instead of directly refining latents in VAE latent space, but refining the VAE encoded latents from other real / generated images.
3. I wonder the relative impact, or the priority of various choices, that is, which plays a bigger role: larger UNet, or each of the conditioning mechanism. I believe these insights are what the research community is insterested in.

---

> ### Author Response · Authors · 2023-11-23
>
> Dear Reviewer pUdr
>
> Thank you for taking the time to review our manuscript and for appreciating our work. We fixed the typos you pointed out.
>
> To answer your questions:
>
> 1. Since reviewer n7kc also asked about the autoencoder, we answer that in our general response. TL;DR: EMA consistently gives better reconstruction metrics. Larger batch sizes provide better gradient updates.
> 2. The refiner can be applied both on real images (in the SDEdit [1] fashion), or to “finish” the denoising process from the base model. Both work equally well, note that the latter option is cheaper when synthesizing from a text prompt. We find that the refiner is also able to add high frequency details when utilizing the SDEdit approach.
> 3. Crop-conditioning in particular can be applied as a finetuning technique and fix issues observed in early versions of SD, where the random cropping augmentation from training leaked into the synthesized samples. However, as we describe in Sec. 2.3, crop-conditioning is mostly beneficial when we train on a fixed resolution, e.g., 512x512 pixels. For the final multi-aspect model, crop-conditioning does not need to be applied. Further, we ablate the effectiveness of size-conditioning in Tab. 2.
>
> References:
>
> [1]: Meng et al: *SDEdit: Guided Image Synthesis and Editing with Stochastic Differential Equations*

---

### Author Response · Authors · 2023-11-23
**General Reply**

We thank all reviewers for taking the time to review our manuscript and the encouraging feedback from the reviews. As both reviewer pUdr and n7kc ask about ablation experiments on the autoencoder, we present them here in a joint answer.

**Autoencoder Ablation:**

To assess the effect of (1) larger batch size and (2) using an exponential moving average (EMA) of the weights, we conduct two experiments, where we train the autoencoder (from scratch) on with (a) a total batch size of 8 and (b) a total batch size of 256. Since we only had limited resources and time available for this rebuttal, these models are not trained until convergence. For both experiments, we evaluate the reconstruction performance on a fixed 10k subset of the COCO2014 validation set, comparing EMA and non-EMA weights for both models. We report rFID, PSNR, SSIM, and LPIPS in a table below.

We see, that

1. In all our evaluations the EMA variants give a consistent improvement compared to the non-EMA variant. This effect is amplified after more update steps (compare A1 and A2)
2. To evaluate the effect of the large batch size we first compare the models when they are both trained for the same number of gradient updates (A1(e) and B(e)): In this case, the large batch size provides a clear benefit in terms of rFID: 7.34 vs 8.85, while the other metrics are relatively close. However, when we evaluate the models after they have trained for roughly the same wall time (500k (B) and 1.9M (A2) update steps), the larger number of updates is a more important factor, with A2e outperforming Be in all metrics.
We believe that training until convergence might change this outcome, since we can see from the B/A1 evaluation that the large batch size provide a better update direction for the weights. This will likely help in finding a loss-minimum.

From this experiment, we conclude that EMA-tracking has a significant effect on our performance, but we believe that the choice of the trade-off between batch size and update steps requires more thorough investigation in a future work.

We thank the reviewers for pointing out this missing piece, and we hope to provide more on this for the research community in the future. For the time we have updated the paper with the concern and added this whole experiment to the appendix.

```jsx
| Model Name  | Batch Size   | EMA   | Global Steps   | rFID [↓]  | PSNR [↑]   | SSIM [↑]  | LPIPS [↓]   |
| ----------- | ------------ | ----- | -------------- | --------- | ---------- | --------- | ----------- |
| B           | 256          | ✗     | 500k           | 7.52      | 24.30      | 0.71      | 1.27        |
| Be          | 256          | ✓     | 500k           | 7.35      | 24.33      | 0.72      | 1.24        |
| A1          | 8            | ✗     | 500k           | 9.14      | 24.42      | 0.72      | 1.32        |
| A1e         | 8            | ✓     | 500k           | 8.85      | 24.51      | 0.72      | 1.29        |
| A2          | 8            | ✗     | 1.9M           | 6.28      | 25.05      | 0.74      | 1.08        |
| A2e         | 8            | ✓     | 1.9M           | 5.57      | 25.05      | 0.74      | 1.07        |
```

---

### Meta-Review · Area_Chair_7mPH · 2023-12-05

**Metareview:**

This paper proposes SDXL for text-to-image synthesis. The main differences to the previous Stable Diffusion are that 1) the network complexity is increased, and 2) various refinement modules are introduced to improve visual quality. The reviewers agree that the proposed SDXL achieves compelling performance and is valuable to the community, but the paper lacks ablation studies on the components. The authors responded to the reviewers in the rebuttal by providing additional experiments. The AC also thinks that the proposed work is valuable, and therefore recommend an acceptance.

**Justification For Why Not Higher Score:**

While SDXL clearly demonstrates its compelling performance in text-to-image synthesis, the analysis of its modules and comparison to other methods are relatively lacking. Moreover, the overall pipeline remains similar to previous methods, although with cleverly designed modules. Therefore, the AC recommends a spotlight.

**Justification For Why Not Lower Score:**

All the reviewers give a high rating (8) to the paper. The AC agrees with the reviewer that the performance is remarkable and it could provide useful insights for future development. Therefore, the AC believes this would be more than a poster paper.

---

### Decision · Program_Chairs · 2024-01-16

Accept (spotlight)